# Exploring the Driving Forces of Vegetation Greening on the Loess Plateau at the County Scale

Chenxiao Kong [1,2,3], Jinghua Huang [1,2,4], Sheng Du [1,2,4] and Guoqing Li [1,2,4,*]

1    The Research Center of Soil and Water Conservation and Ecological Environment, Chinese Academy of Sciences and Ministry of Education, Yangling, Xianyang 712100, China; kongchenxiao21@mails.ucas.ac.cn (C.K.); jhhuang@nwsuaf.edu.cn (J.H.); shengdu@ms.iswc.ac.cn (S.D.)
2    Institute of Soil and Water Conservation, Chinese Academy of Sciences and Ministry of Water Resources, Yangling, Xianyang 712100, China
3    University of Chinese Academy of Sciences, Beijing 100049, China
4    State Key Laboratory of Soil Erosion and Dryland Farming on the Loess Plateau, Northwest A&F University, Yangling, Xianyang 712100, China
*    Correspondence: liguoqing@nwsuaf.edu.cn; Tel.: +86-29-8701-2411

**Abstract:** China has implemented several ecological projects in the Loess Plateau region to address severe land degradation and soil erosion. Accurately assessing ecological restoration and its driving factors remains challenging. Previous studies in this area concentrated on driving factors have mainly focused on natural factors at the regional or watershed scale, with limited consideration of socioeconomic factors at the county scale. In this study conducted in Huanglong County on the Loess Plateau, the focus was to fill the gaps in previous research and provide insights into the socioeconomic driving forces behind vegetation greening. Remote sensing image data (NDVI) from 1999 to 2019 were used to analyze vegetation greenness dynamics in the region. Five socioeconomic variables were considered, including afforestation intensity, deforestation intensity, agricultural intensity, village intensity, and road intensity layers, to characterize the impact of afforestation, agriculture, and urbanization policies. The RESTREND (residual trends) method was employed to assess the relative importance of climate and human activities on vegetation dynamics. This study found that temperature–NDVI relationships are more suitable for building RESTREND models than precipitation–NDVI relationships. Human activity was the main driver of vegetation dynamics, contributing 62% compared to 38% from climate change. Agricultural practices and afforestation were found to have a positive impact on ecological restoration, while deforestation and urbanization had no significant impact. These findings highlight a conceptual framework for understanding the intricate relationship between ecological restoration, climatic factors, and human activity on the Loess Plateau. This study suggests that significant progress has been made in ecological restoration through human efforts in combating land degradation. However, it emphasizes the need to strengthen natural conservation efforts and gradually transition toward restoration processes driven by natural forces for sustainable socioeconomic development. The methodology used in this study can be applied to explore the driving forces of ecological restoration in other regions facing human-driven land degradation.

**Keywords:** land degradation; ecological restoration; driving force analysis; RESTREND; contribution rate



## 1. Introduction

Land degradation is a significant issue affecting a large portion of the global land area, particularly in dry areas. According to estimates from the Global Environmental Fund, if the current trend continues, it is projected that by 2050, a significant portion of the world's land area, approximately 95%, will become degraded. This has serious implications for the approximately 3.2 billion people who live in these areas, as well as for the overall

health of the planet [1]. Efforts to combat land degradation and implement sustainable land management practices are crucial to mitigate these negative impacts.

Land degradation refers to the deterioration of land quality and productivity caused by various processes, including soil erosion, salinization, loss of soil fertility, and depletion of seed banks [2]. It can occur in arid, semi-arid, or subhumid areas [3]. Both climate variations, such as high temperatures, low annual rainfall, and high interannual variability, as well as human activities, are the driving forces behind land degradation [4–7]. Human-induced degradation is often cited as a primary cause of desertification in semi-arid areas [8,9]. Overgrazing, agricultural development, creation or movement of human settlements, and unsustainable land-use practices are among the main contributors to desertification [10–12]. Land degradation has significant impacts on both the biological and economic productivity of an area [13–15]. It affects vegetation cover, biodiversity, and density, leading to a loss of ecosystem services. Additionally, it affects agricultural productivity, food security, and the livelihoods of local communities [16,17].

Ecological restoration plays a crucial role in combating land degradation by improving soil quality, enhancing biodiversity, and promoting sustainable land management practices, especially in semi-humid and semi-arid areas [18,19]. It involves rehabilitating natural sites that have been degraded or damaged by human activities through measures like reforestation, soil conservation, and habitat restoration [20]. The goal is to return these sites to their previous state or a condition closely linked to the one unaltered by human influence [21]. The relationship between ecological restoration and land degradation is that ecological restoration aims to reverse or mitigate the negative impacts of land degradation. International organizations, such as the United Nations Convention to Combat Desertification (UNCCD), are actively involved in addressing land degradation. The UNCCD promotes cooperation in combating desertification and mitigating the effects of drought. Additionally, the Food and Agriculture Organization (FAO) and other environmental agencies collaborate to raise awareness, develop sustainable land management practices, and implement rehabilitation measures in affected regions. More information about these organizations and their efforts could be found on their respective websites: UNCCD (https://www.unccd.int/, accessed on 1 March 2024) and FAO (https://www.fao.org/, accessed on 1 March 2024).

Evaluating the effectiveness of ecological restoration in controlling land degradation requires the use of various indicators. These indicators include changes in vegetation cover/greenness, soil erosion rates, soil fertility, biodiversity loss, and water quality. These indicators vary in terms of data collection, analysis, and interpretation [22]. They can be categorized into three main types: remote-sensing-based techniques, field-survey-based techniques, and socioeconomic-based techniques.

Remote-sensing-based techniques utilize satellite imagery and other remote sensing technologies to assess changes in land cover, vegetation health, and soil conditions [23,24]. These technologies offer broad-scale and continuous monitoring of land degradation over large areas. They can detect changes in vegetation cover, soil erosion, and land-use patterns. Remote sensing techniques provide efficient data on a regional or global scale. However, it is important to note that they may not provide detailed information on specific land-use practices and socioeconomic factors [25].

Field-survey-based technologies involve on-site data collection and analysis. This method includes field observations, soil sampling, and vegetation assessments. Field surveys provide detailed information on specific locations and allow for a direct assessment of land degradation indicators, such as soil erosion rates and vegetation health [26]. They can also capture local variations in land-use practices and socioeconomic factors. However, field surveys are time-consuming and resource-intensive, making them less efficient for large-scale monitoring.

Socioeconomic-based technologies focus on the analysis of socioeconomic indicators and data related to land use, agriculture, and human activities [27]. These indicators can include changes in agricultural practices, land-use patterns, population density, and economic factors. Socioeconomic-based technologies provide insights into the human

drivers and impacts of land degradation. They help us to understand the underlying causes of degradation and can guide policy and decision-making processes [28,29]. However, it is important to note that these technologies may be limited in capturing the physical changes in land cover and soil conditions.

Each monitoring approach has its advantages and limitations. Remote-sensing-based technologies provide a broad-scale overview, field-survey-based technologies offer detailed on-the-ground information, and socioeconomic-based technologies provide insights into the human dimensions of land degradation. For large-scale and spatially patterned assessments, remote-sensing-based methods have strong advantages over the other two methods.

China has indeed achieved remarkable ecological restoration from land degradation over the past three decades [30], particularly in the Loess Plateau [31–33]. The government has implemented various ecological engineering and conservation policies, such as returning farmland to forest projects, protecting natural forests, and prohibiting grazing and logging [34]. These policies, supported with substantial funding, have resulted in positive outcomes, including vegetation recovery, effective soil erosion control, and significant social and economic development [35–37]. However, there is an ongoing discussion among scholars regarding the extent to which human activities contribute to vegetation greening on the Loess Plateau [38–40]. Most research has focused on vegetation greenness at the regional [38,41–45] or watershed scale [46–50], with limited studies conducted at the county scale. Additionally, the analysis of driving factors behind spatiotemporal changes in vegetation greenness has mainly focused on natural factors [51–53], with limited consideration of socioeconomic factors.

The Loess Plateau is composed of over 340 counties [54], which serve as the basic administrative units in China. Counties have independent operational characteristics and diverse regional functions [55], making them crucial in implementing ecological initiatives, safeguarding natural forests, and conducting reforestation programs [56]. Huanglong County, located in the central Loess Plateau, has experienced significant changes in land use and cover over the past two decades due to various ecological initiatives. It serves as a representative area for ecological governance and restoration. The aim of our study was to analyze satellite imagery data for Huanglong County over a period of 21 years in order to address the research gap. Our specific objectives were as follows:

(1)  Determine the trend of changes in vegetation greenness;
(2)  Identify the dominant factor, whether climate change or human activities, influencing the county-scale ecological restoration progress;
(3)  Explore the human dimension that affects ecological restoration.

Through this study, we aimed to provide an accurate and comprehensive assessment of the processes of vegetation greening and the underlying causes of restoration. This research will contribute to the development of a scientifically sound ecological restoration framework at the county level, helping combat land degradation in a meaningful way.

## 2. Materials and Methods

### 2.1. Study Area

Huanglong County (35°24′09″–36°02′11″ N, 109°38′49″–110°16′49″ E), located in Shaanxi Province, China, is part of the Loess Plateau (Figure 1). It covers an area of 2751 km$^2$ and has hilly terrain with elevations ranging from 656 m to 1774 m. The county has a distinct continental climate, with an average annual temperature of 9.51 °C and an average annual rainfall of 582 mm, mainly occurring between July and September (Figure 2).

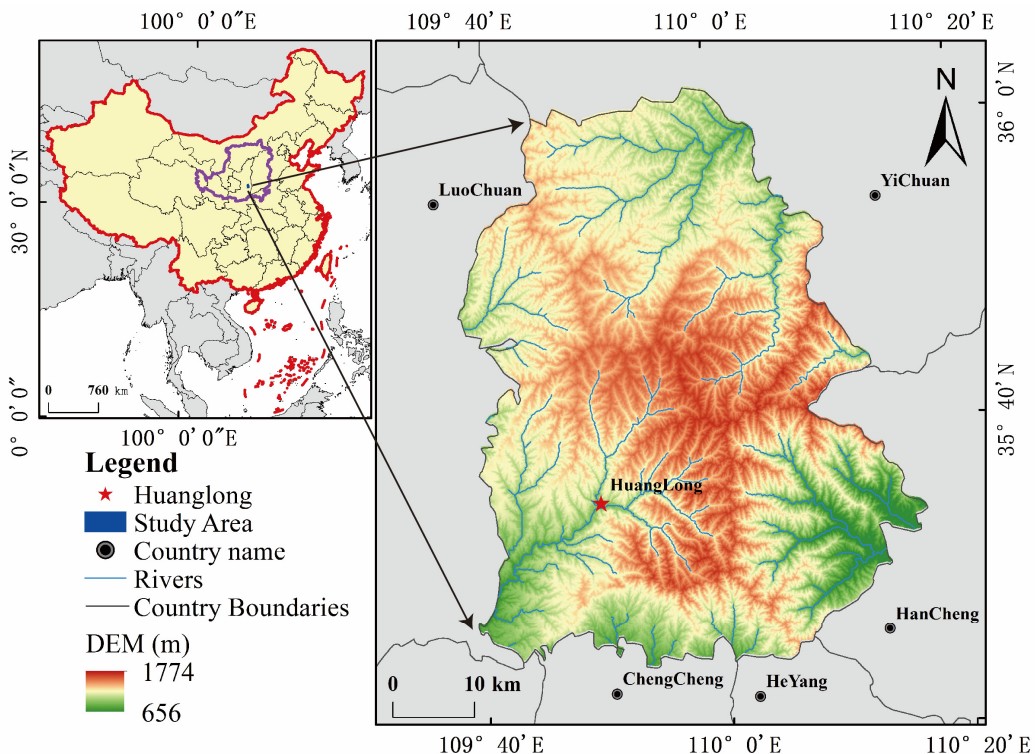

**Figure 1.** The location of Huanglong County on the Loess Plateau of China.

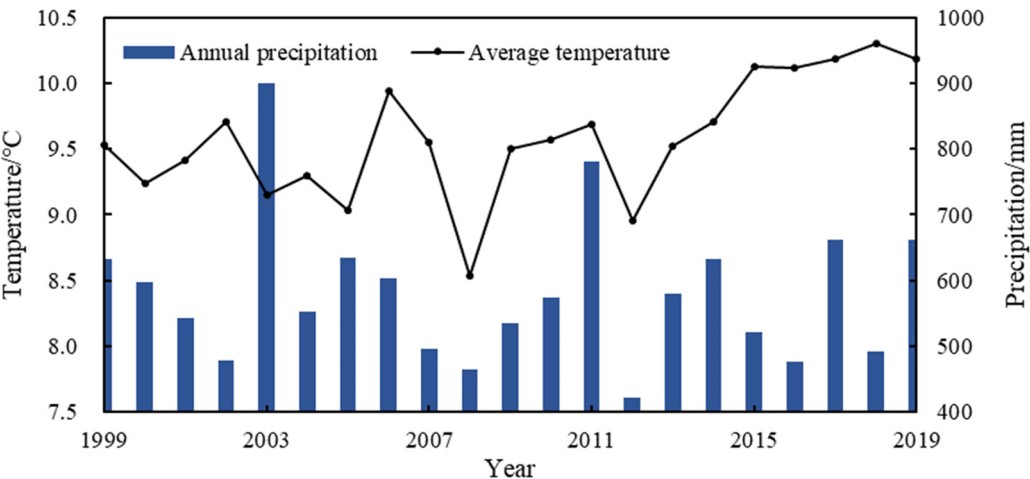

**Figure 2.** Precipitation and temperature in Huanglong County from 1999 to 2019.

In terms of forest, Huanglong County encompasses various types, including evergreen broad-leaved forest, bottom mountain secondary shrubland, coniferous and broad-leaved mixed forest, and coniferous forest. The main natural forest species in the county include *Quercus wutaishanica*, *Pinus tabulaeformis*, and *Robinia pseudoacacia*. On the other hand, apples, peaches, and apricots are the dominant economic forest species.

The county has a population density of 14.54 individuals per square kilometer and an urbanization rate of 57.12% (2020). It has leveraged its abundant natural resources and well-developed road transportation to develop specialty industries such as walnuts, honey, and hairy crabs, contributing to an annual GDP of 2.305 billion.

Due to extensive exploitation of natural resources, agricultural activities, grazing, and deforestation, Huanglong County has been facing severe soil erosion. In response, ecological engineering projects, including the "Grain for Green" program and the Natural Forest Protection Project, have been implemented since the 1990s. Protective policies such

as mountain closure and nature reserve construction have also been put in place to address these issues.

*2.2. Data Sources*

2.2.1. Remote Sensing Data

The SPOT satellite (Satellite Pour I' Observation de la Terre), developed by the Centre National D'Etudes Spatiales (CNES, Paris, France), is an earth observation satellite used for remote sensing. Since 1986, a series of SPOT satellites (SPOT 1–7) have been launched, providing continuous remote sensing data for various application fields. The SPOT satellite can capture images in the visible light band as well as capturing multispectral information, including infrared bands. This makes the SPOT satellite particularly valuable for analyzing vegetation, land cover, water bodies, soil, and other related studies. In vegetation research, SPOT satellite images can be used to obtain vegetation coverage information, monitor changes in vegetation, assess vegetation health, and more. Due to its high spatial resolution and multispectral capabilities, the SPOT satellite is considered an important data source in remote sensing research.

Vegetation indices, such as the Normalized Difference Vegetation Index (NDVI), are commonly used in remote sensing to assess vegetation health and abundance [57]. The NDVI is calculated using the reflectance values of near-infrared (NIR) and red (R) wavelengths from satellite imagery [58]. It provides valuable information about vegetation cover and its changes over time. The value of the NDVI typically ranges from 0 to 1, with larger values indicating thriving vegetation. By comparing these two spectral bands, the NDVI can provide insights into vegetation density, growth stages, and overall health.

The vegetation index, particularly the NDVI, is widely used in various applications such as agriculture, forestry, environmental monitoring, and land degradation assessment [59,60]. It helps monitor changes in vegetation cover, identify areas affected by drought, analyze land-use patterns, and assess the impact of human activities on ecosystems [61]. In agriculture, the NDVI can be used to monitor crop health [62], identify growth cycles, support irrigation and fertilization plans [59], and promote sustainable agricultural management [63]. It provides valuable information for planning and protecting agricultural resources. Overall, the vegetation index, especially the NDVI, is a valuable tool for understanding vegetation dynamics and monitoring environmental changes over time.

A continuous annual maximum NDVI dataset for the Loess Plateau, covering the period from 1999 to 2019, was gained from the Resource and Environment Science and Data Center (RESDC, https://doi.org/10.12078/2018060601, accessed on 1 February 2023) [64]. This dataset consists of 21 layers of maximum NDVI values derived from SPOT/VEGETATION. The maximum synthesis method was used to mitigate the influence of cloud cover and other obstructions, capturing the peak value of vegetation during the growing season. The dataset has a temporal resolution of 1 year, a spatial resolution of 1 km $\times$ 1 km, and uses the Albers_Conic_Equal_Area coordinate projection. This dataset is widely used for monitoring vegetation changes at various scales and is particularly valuable for studying vegetation change trends in larger areas [65,66]. While the 1 km resolution may not capture microscopic changes in vegetation, it is sufficient for identifying and analyzing the main patterns of vegetation change, especially at the county scale.

2.2.2. Climate Data

The climate data used in this study were obtained from the China Meteorological Data Service Center for the years from 1999 to 2019 (http://data.cma.cn, accessed on 1 February 2023). The data were directly monitored at the meteorological station in Huanglong County. The precipitation in Huanglong County ranged from 422.5 mm to 899.4 mm, with an annual average temperature fluctuating between 8.5 °C and 10.3 °C. Since Huanglong County has only one meteorological station, it is assumed that the precipitation and temperature data are uniformly distributed and there is no spatial heterogeneity across the county. This assumption is reasonable considering that the minimum level for establishing a

national meteorological station in China is at the county level. It is important to note that the climate data in each grid cell of Huanglong County are identical to the data collected at the meteorological station. The meteorological stations in the surrounding counties are far away and not representative for Huanglong county, which may result in great errors for the interpolation of climate layers based on surrounding meteorological stations [67]. Furthermore, the meteorological data obtained for this study are based on monthly observations.

2.2.3. Socioeconomic Factors

Socioeconomic factors considered in this study include afforestation intensity, deforestation intensity, agricultural intensity, village intensity, and road intensity layers. These factors represent three types of socio-economic policies: forestry, agriculture, and urbanization.

The forestry policy comprises afforestation intensity and deforestation intensity layers. The afforestation intensity layer is created using a kernel density estimation function, utilizing a binary layer that transitions from nonforest land-use type to forest land-use type between 1999 and 2019. On the other hand, the deforestation intensity layer is generated with a similar method, but for the transition from forest land-use type to nonforest land-use type during the same period. The agriculture policy involves the agricultural intensity layer, which is produced using a kernel density estimation function based on cropland land-use type data from 2019. For the urbanization policy, village intensity and road intensity layers are considered. The village intensity layer is created through a kernel density estimation function using village points data from 2019. Similarly, the road intensity layer is generated using a kernel density estimation function based on road alignment data from 2019.

The binary layer transitions between nonforest land-use type and forest land-use type during 1999 and 2019 were created using the land-use data from those respective years. Similarly, the binary layer transitions from forest land-use type to nonforest land-use type during 1999 and 2019 were generated by comparing the land-use data of those years. The binary layer representing cropland land-use type in 2019 was produced based on the land-use data from that specific year. In the generation of afforestation intensity and deforestation intensity data, the comparison of land-use conditions in 1999 and 2019 was conducted to capture the changing trends in the region over the 21-year period. While this approach does not provide detailed annual change information, it offers insights into the long-term trends and overall directions of change in the area. Regarding the data on agricultural intensity, village intensity, and road intensity, the selection of data from 2019 allows for the assessment of the current impact of agricultural activities and urbanization levels on vegetation restoration in the region.

The land-use data for 1999 and 2019 were sourced from Yang and Huang (2021), with an overall accuracy of 79.31% [68]. Road maps and village points were acquired from the OpenStreetMap database (https://download.geofabrik.de, accessed on 1 February 2023). The kernel density estimation function is implemented using ArcGIS10.6 (Environmental Systems Research Institute, Redlands, CA, USA). To normalize values between 0 and 1, the intensity layers were scaled using the formula (X-cellStats(X,min))/(cellStats(X,max) − cellStats(X,min)) using the RASTER package in R4.3 (R Foundation for Statistical Computing, Vienna, Austria). The driving factor layers used in the analysis are depicted in Figure 3.

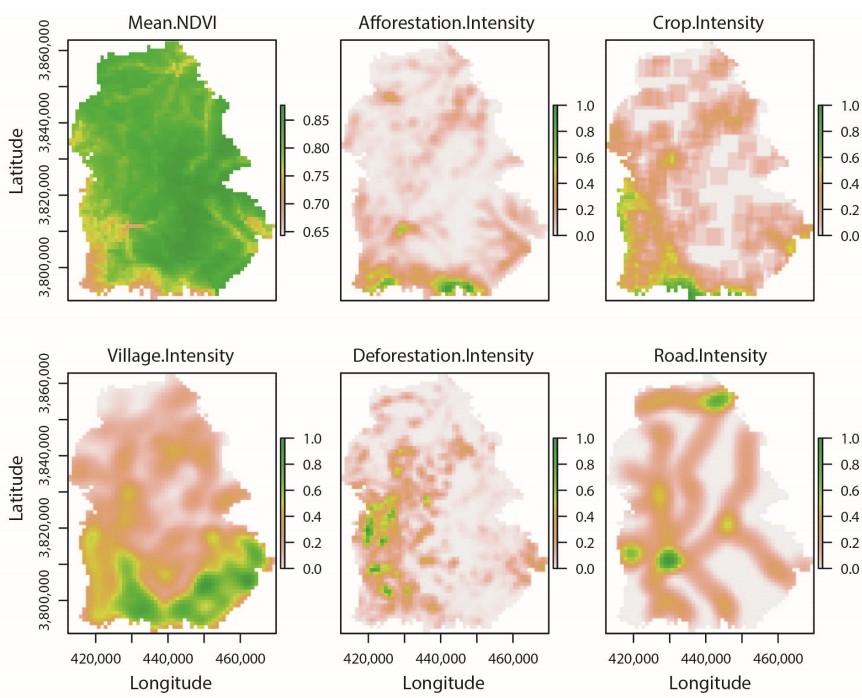

**Figure 3.** Map of annual average of maximum NDVI (1999–2019) and five socioeconomic factors representing human activities.

## 2.3. Experimental Design and Data Statistical Analysis

### 2.3.1. Residual Trends Method

This study utilized the residual trends (RESTREND) analysis method to investigate the factors driving NDVI change in Huanglong County. The RESTREND method, first used by Evans and Geerken [69], is considered as a reliable method for distinguishing the impacts of climate- and human-accelerated land degradation [70]. The methodology controls for climate variations by establishing a regression between the NDVI (a measure of vegetation greenness or ecosystem productivity) and precipitation [71–73]. The observed NDVI values subtract the simulated NDVI values estimated from the regression model, the results are known as NDVI residuals. A trend analysis is applied on the residuals, with a positive trend indicating ecological restoration [74–76]. Further information about the RESTREND method can be seen in Wessels et al. [71] and Evans and Geerken [69].

The method described in this study is based on two assumptions. Firstly, it establishes a relationship between climate factors and NDVI to obtain predicted values, which represent the variability directly attributed to climate factors. Secondly, the residual is calculated as the difference between actual observed values and predicted values, capturing the variation that cannot be explained by climate alone. Typically, this unexplained variation is attributed to human activities. It is important to note that the essence of RESTREND thought is binary division and does not account for the influence of random processes. By utilizing predicted values and residuals, it becomes possible to effectively distinguish the impact of climate change and human activities on NDVI changes.

### 2.3.2. The Workflow for Statistical Analysis

RESTREND analysis has typically been developed to identify land degradation in dryland grassland or desert ecosystems, primarily considering precipitation as the main influencing factor [74]. However, it is important to note that this study conducted in Huanglong County is situated in a semi-humid forest region, where temperature may also play a significant role in driving vegetation greenness changes. Therefore, in constructing the regression equation at the grid-cell scale, this study incorporates both precipitation and

temperature factors. The key steps involved in utilizing this method to monitor long-term ecological restoration of vegetation index are as follows.

Firstly, the 21 *NDVI* raster data for Huanglong County were converted into 21 vector data using R language, resulting in an *NDVI* matrix with dimensions of 2811 × 21 [grid × layer]. Since vegetation growth in the Loess Plateau can be influenced by both temperature and precipitation [77], this study adopted a regression analysis model that considers the combined effects of temperature and precipitation. For each grid, a multivariate stepwise regression analysis was performed to establish a regression equation, which can be formulated as follows:

$$Obj_i = step\left(lm\left(NDVI_{[i,]} \sim gp + gt + gp^2 + gt^2\right)\right) \quad (1)$$

where $Obj_i$ represents the optimal equation for grid $i$, $NDVI_{[i,]}$ represents the 21-year sequence of *NDVI* data for grid $i$; and $gp$ and $gt$ represent the 21-year growing season total precipitation and mean temperature data from the meteorological stations, respectively. Additionally, $gp^2$ and $gt^2$ are the squares of the corresponding values of growing season total precipitation and mean temperature, respectively. This equation is established to investigate the nonlinear relationship between climatic factors and the *NDVI*.

Secondly, the expected values for each grid are predicted using optimal multivariate equations specific to each grid. The regression residual is calculated as the difference between the observed real values and the expected values. The formula can be expressed as follows:

$$Resd_i = NDVI_{[i,]} - predict(Obj_i) \quad (2)$$

where $NDVI_{[i,]}$ represents the 21-year sequence of the *NDVI* data for grid $i$, $predict(Obj_i)$ is the predicted value of the *NDVI* for each grid, and $Resd_i$ is the residual between the real $NDVI_{[i,]}$ and $predict(Obj_i)$ for each grid.

Thirdly, a linear regression equation was employed to calculate the 21-year regression residuals for each grid and corresponding year. The formula used for this calculation is as follows:

$$Tred_i = coef(lm(Resd_i \sim year)) \quad (3)$$

where $Tred_i$ represents the residual trend (regression slope) for grid $i$, and *year* marks the duration from 1999 to 2019. The *coef* represents a function for extracting regression slopes. When $Tred_i > 0$, human activities had a positive effect on ecological restoration, while $Tred_i < 0$ indicates a negative impact.

Fourthly, the relative importance of human activity and climate change on vegetation greenness can be calculated using the following formula:

$$Ac = diff(range(predict(Obj_i))) \quad (4)$$

$$Ah = diff(range(Resd_i)) \quad (5)$$

$$C_{h_i} = \frac{Ah}{Ac + Ah} \quad (6)$$

In these formulas, *Ac* indicates the amplitude of changes in the *NDVI* caused by climate change; *Ah* indicates the amplitude of changes in the *NDVI* caused by human factors; $C_{c_i}$ characterizes the rate of climatic contribution to the *NDVI* dynamics; and $C_{h_i}$ characterizes the rate of human activity contribution to the *NDVI* dynamics. The *diff* represents a function for calculating the difference between two data points. The *range* represents a function for calculating the range size of a vector.

Finally, we investigated the impact of human activities on regression residuals. Here, we calculated the correlation between the *NDVI* and each socioeconomic variable using the Pearson correlation coefficient. The formula used for these calculations is as follows:

$$R_i = cor(Trend, social[[i]], method = "pearson") \quad (7)$$

where *social*$[[i]]$ is the *i* human disturbance factor (afforestation intensity, deforestation intensity, agricultural intensity, village intensity, and road intensity), and $R_i$ is the correlation coefficient between *Trend* and the *social*$[[i]]$. The value range of *R*, the correlation coefficient, is between $-1$ and 1. Positive values indicate a positive correlation, while negative values indicate a negative correlation. The absolute value of the correlation coefficient is used to determine the strength of the correlation. In general, a range of 0–0.1 indicates a negligible correlation, a range of 0.1–0.39 indicates a weak correlation, a range of 0.4–0.69 indicates a moderate correlation, a range of 0.7–0.89 indicates a strong correlation, and a range of 0.9–1 indicates a very strong correlation [78]. To test the hypothesis, the threshold of moderate correlation was used to detect the important social factor that affects residual trends. Variables with *R* greater than this threshold will be considered the main factors affecting residual trends.

All the analyses were carried out using the R4.3 software package (https://www.r-project.org/, accessed on 1 February 2023) and ArcGIS 10.6 (Environmental Systems Research Institute, Redlands, CA, USA). The overall workflow is shown in Figure 4.

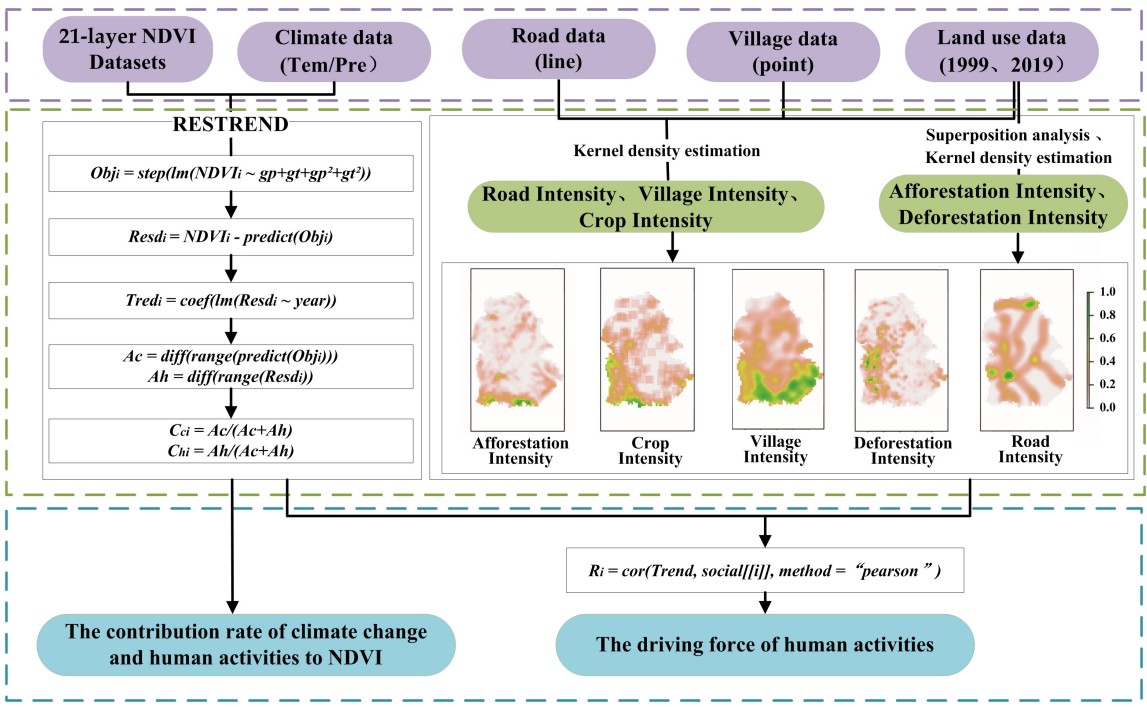

**Figure 4.** The workflow for studying the driving forces of vegetation greening.

## 3. Results

### 3.1. Stepwise Regression and Residual Trend Analysis

We extracted grid values and climate data from the 21-year raster layer at corresponding locations. Each NDVI grid value was considered as the dependent variable, while the climate data were treated as the independent variable in establishing a regression relationship. This process was repeated for 2811 grids. Table 1 provides a summary of the regression results, indicating the established correlation between the NDVI and climate factors in our study. The table shows that 95% of the grids exhibited good predictive performance. Most regression models established on grids are linear with respect to temperature rather than to precipitation. This means that temperature variation has a greater impact than precipitation on the dynamics of vegetation greenness. However, the model also showed poor prediction performance for approximately 5% of the grids distributed in the western part, indicating that their vegetation greenness may not be affected by precipitation or temperature. Additionally, the overall $R^2$ value of the stepwise regression based on the

grids was found to be significantly high (Figure 5A–D). From a spatial perspective, the model showed better predictive performance for grids in the northern region.

**Table 1.** Statistics on the relationship between the NDVI and growing season precipitation (*gp*)/temperature (*gt*) at the grid scale.

| Formula | Grids | Percentage | NDVI–Climate Relationship |
|---|---|---|---|
| *NDVI ~ gt* | 2671 | 95.0% | NDVI–temperature linear relationship |
| *NDVI ~ 1* | 137 | 4.9% | No NDVI–climate relationship |
| *NDVI ~ gt + gp + gt² + gp²* | 3 | 0.1% | NDVI–climate nonlinear relationship |
| Total | 2811 | 100% | |

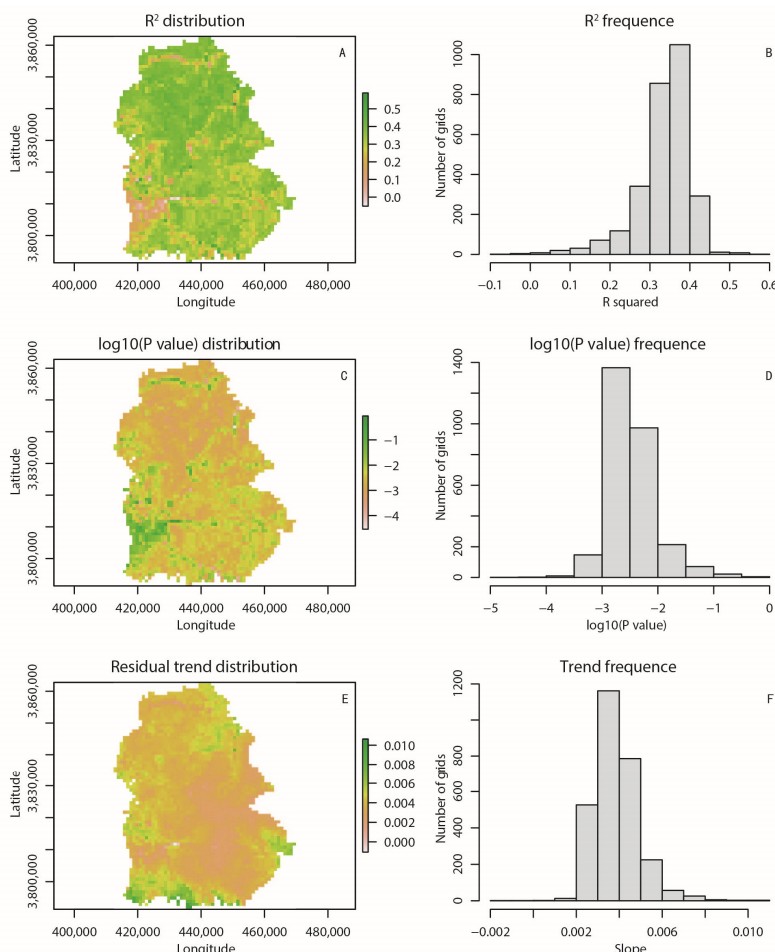

**Figure 5.** The spatial pattern (**A**) and frequency distribution (**B**) of the regression R² value (climate–NDVI relationship, calculated with Equation (1)), and the spatial pattern (**C**) and frequency distribution (**D**) of the regression *p* value (climate–NDVI relationship, calculated with Equation (1)), as well as the spatial pattern (**E**) and frequency distribution (**F**) of the residual trend (regression slope, residual–year relationship, calculated with Equation (3)) between the NDVI and climate over 1999–2019. The *p*-value undergoes a conversion using log10 for the purpose of displaying it in a normalized manner.

We visualized the spatial distribution of the residual trend (regression slope) for 2811 grids using a raster layer, as shown in Figure 5E. In the figure, a positive value represents a positive trend. This two-dimensional representation allows readers to perceive the changes in trends more effectively compared to a line chart with a single value. Figure 5F provides a statistical indicator that helps us understand the frequency of the residual trend in the regression equation. It indicates that 98% of the grid values were greater than 0,

indicating a positive residual trend. This suggests a positive impact of human activities on ecological restoration from 1999 to 2019. The higher values of the residual trends were primarily concentrated in the southwestern region of Huanglong County.

### 3.2. Climate and Human Contributions

The decomposition of variation reveals that human activities have emerged as the primary driving force behind vegetation change, accounting for 62% of the recovery, while climatic factors have contributed 38% (Figure 6A). This indicates that human interference has a much larger impact on vegetation greenness compared to climate change.

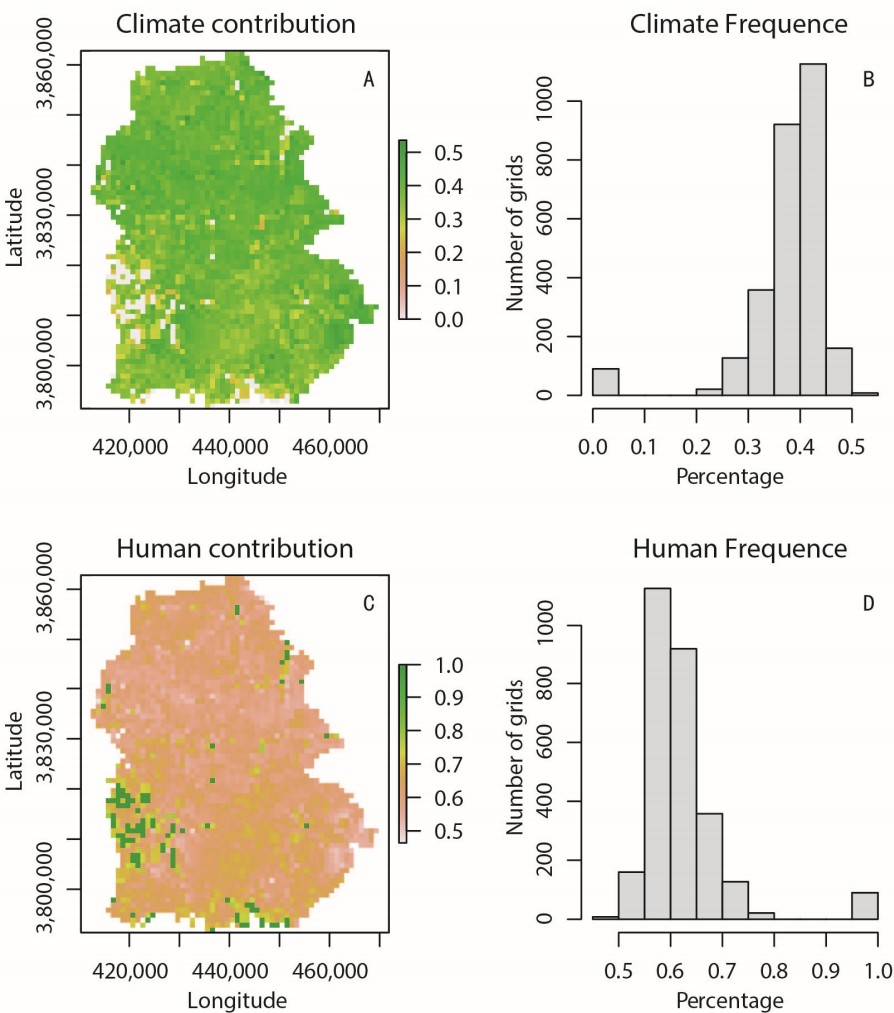

**Figure 6.** The spatial pattern (**A**) and frequency distribution (**B**) of the climate contribution, as well as the spatial pattern (**C**) and frequency distribution (**D**) of the human contribution to NDVI change over 1999–2019.

When examining the spatial distribution of these contributions within the region, it becomes apparent that the western parts are more influenced by human activity. This suggests that human-induced land-use practices and ecological management in this area have had a significantly positive effect on vegetation conditions. These positive influences may include more intensive land-use practices and urban greening initiatives, among other socioeconomic factors. Overall, these efforts have collectively facilitated the restoration of vegetation and ecosystems.

### 3.3. The Driving Force of Human Activities

The analysis in Figure 7 reveals that there is a positive correlation between residual trends and all socioeconomic factors. Among these factors, we did not find strong or very strong correlations among these 5 factors (*R* value greater than 0.7). However, afforestation and agriculture have moderate correlation levels, with absolute values of *R* equal to 0.4 and 0.59, respectively. This suggests that afforestation density and agricultural density play significant roles in the socioeconomic factors affecting vegetation recovery. On the other hand, village density, deforestation density, and road density show weak or negligible correlations with residual trends. This indicates that forest logging and urbanization have no significant impact on ecological restoration.

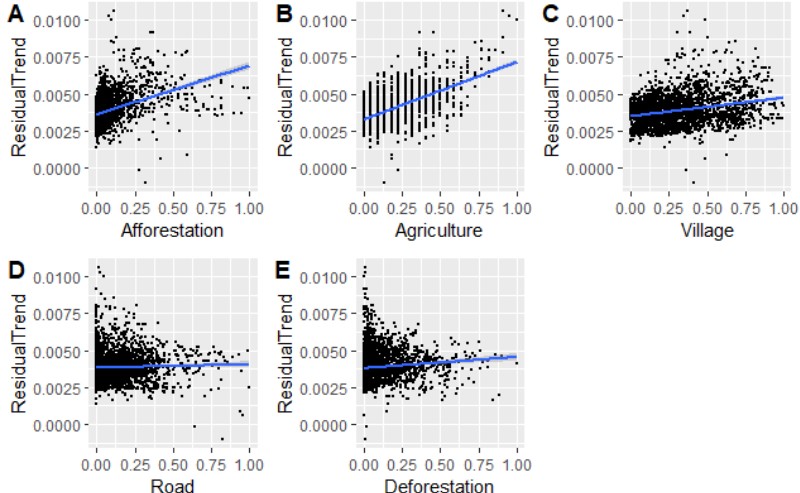

**Figure 7.** The correlation between residual trends and socioeconomic factors. The correlation coefficient is equal to (**A**) 0.4 for afforestation intensity, (**B**) 0.59 for agriculture intensity, (**C**) 0.27 for village intensity, (**D**) 0.03 for road intensity, and (**E**) 0.1 for deforestation intensity.

## 4. Discussion

### 4.1. Precipitation–NDVI vs. Temperature–NDVI Relationship

This study investigated the driving forces behind ecological restoration at the county scale within the Loess Plateau region. Our research findings suggest that there is a distinct linear correlation between vegetation and temperature, indicating that temperature plays a predominant role in the vegetation–climate relationship, surpassing the influence of precipitation. This finding differs from prior studies that focused primarily on precipitation factors in arid desert and grassland regions, assuming precipitation to be the main driving force behind vegetation change [74]. For example, Wang et al. analyzed the spatial patterns of the NDVI in response to precipitation and temperature in the central Great Plains and found relatively high correlations between accumulated precipitation and the NDVI, suggesting that precipitation plays a significant role in vegetation dynamics [79]. Another study by Yang et al. assessed the relationship between AVHRR/NDVI and climatology in Nebraska, USA, and observed a higher correlation between precipitation and the NDVI compared to temperature [80]. However, it is important to carefully consider whether this strong correlation between vegetation and precipitation still holds true in humid forest areas rather than dryland grassland/desert regions, as this aspect has been rarely addressed in previous research.

In general, vegetation in humid regions tends to be more sensitive to temperature variations, while vegetation in arid regions is more sensitive to rainfall variation. This is because, in humid regions where rainfall is abundant, water availability is usually not a limiting factor for plant growth [37]. Therefore, changes in rainfall cannot directly affect vegetation growth and productivity. On the other hand, in arid regions, where water is scarce, vegetation relies heavily on the limited rainfall it receives. Even small changes

in rainfall can have a significant impact on plant growth and survival. Temperature, in contrast, plays a crucial role in determining the rate of plant physiological processes, such as photosynthesis and evapotranspiration. Therefore, the vegetation in humid regions with cold temperature, like Huanglong County, is more sensitive to temperature variations as it directly affects their metabolic activities and water loss through transpiration. Zhan et al. also noted a positive correlation between the annual NDVI and temperature in semi-humid areas [81]. This means that selecting the types of climate factors is crucial when using the RESDTRED method to detect driving forces of change in forests in humid areas. The traditional precipitation–NDVI relationship in grassland/desert regions may not be applicable to forest regions. The inclusion of temperature factors in local regression relationships is necessary to make them applicable to forest ecosystems. However, the presence of time lags may introduce uncertainty, as mentioned in Section 4.3 of this paper. This uncertainty has implications for the analysis and interpretation of the data.

*4.2. Human Dimensions on Vegetation Greening*

Our study suggests that human activities have emerged as the primary driving force behind vegetation change, contributing 62% to the recovery of vegetation, while climatic factors only accounted for 38%. This indicates that the impact of human interference on vegetation greenness far outweighs that of climate change. It is worth noting that our findings differ slightly from broader research on the Loess Plateau, which emphasizes the approximately equal importance of human activities and climatic factors in the process of vegetation restoration. For example, Shi et al. indicated that the contribution of human factors to vegetation growth was 54.2% and that of climatic factors was 45.8% [82].

Furthermore, our study has incorporated more empirical evidence on human activity factors than previous studies, including afforestation, deforestation, and agricultural and urbanization policies. Our findings reveal that afforestation and agricultural policies have positively influenced vegetation recovery and enhanced vegetation restoration. In contrast, deforestation and urbanization have shown negligible impacts on vegetation dynamics. Based on our study, we have developed a conceptual model (Figure 8) to illustrate the vegetation changes in Huanglong County. This framework model reflects deliberate and systematic human interventions aimed at the restoration or reconstruction of ecosystems that have been subjected to natural or human-induced disturbances, degradation, or damage, with the goal of facilitating their gradual return to their original state, in line with the definition of ecological restoration.

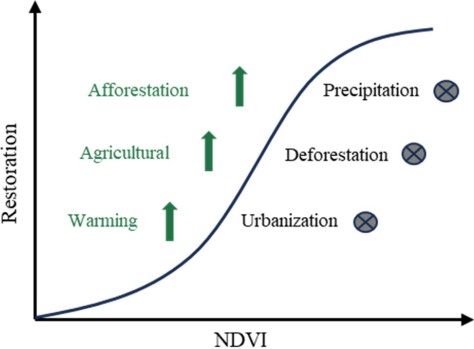

**Figure 8.** A conceptual framework for analyzing ecological restoration in Huanglong County. It shows that afforestation and agricultural policies have played a significant role in enhancing ecological restoration in the region. On the other hand, the impacts of deforestation and urbanization on ecological restoration have been negligible. Interestingly, temperature, rather than precipitation, significantly impacts ecological restoration.

### 4.2.1. Afforestation Policies

Afforestation policies, such as the Grain for Green program, have played a significant role in enhancing vegetation restoration in Huanglong County. The Grain for Green program is a notable afforestation policy in China that promotes vegetation greening through various measures and strategies. These policies focus on restoring degraded forests and establishing new forest areas. Measures include identifying steep slope farmlands (>25°), providing state funding to incentivize farmers to retreat from these areas, and implementing manual or unmanned aerial vehicle planting of seedlings or grass seeds [83].

For example, during the first phase of the afforestation program from 1999 to 2012, Huanglong County experienced significant land-use transformations. This involved allocating 4220 hectares for afforestation on previously cultivated land, reforesting 2907 hectares of barren hills, and implementing forest conservation and nurturing initiatives on 800 hectares of mountainous terrain. As a result of these afforestation efforts, vegetation cover, vegetation greening, and ecosystem functions have increased in the county.

### 4.2.2. Agricultural Policies

Agricultural policies have indeed played a role in enhancing vegetation restoration in Huanglong County. China has implemented several agricultural improvement policies. These policies aim to protect farmland and promote the development of high-standard farmland. They include measures such as prohibiting the occupation of farmland, reshaping the agricultural land surface, upgrading agricultural infrastructure, using machinery for seed sowing, building irrigation systems, applying scientific fertilization techniques, and developing advanced planting technology. These activities create favorable conditions for vegetation growth and help compensate for limited rainfall in dryland areas through adequate water supply [84].

Furthermore, agricultural practices such as crop rotation, soil conservation, and nutrient management can improve soil fertility and promote healthy plant growth [85]. However, it is important to note that the expansion of agriculture may have negative consequences, such as the conversion of natural vegetation into agricultural land, leading to a loss of biodiversity and disruptions in the natural ecosystem [86]. Therefore, implementing sustainable agricultural practices is crucial to minimize negative impacts and ensure long-term vegetation greening. Overall, agriculture contributes to vegetation greening by improving water availability, nutrient supply, and proper land management practices. However, it is essential to carefully consider the ecological balance and implement sustainable practices to avoid detrimental effects on the environment.

### 4.2.3. Deforestation and Urbanization Policies

Usually, deforestation and urbanization are not conducive to vegetation greening [87]. However, our study indicates that in the case of Huanglong County, deforestation and urbanization have had negligible impacts on vegetation greening. We speculate that this is due to the implementation of China's National Forest Conservation Program and national urban greening policies, which aim to protect vegetation from adverse factors.

China's National Forest Conservation Program aims to address the issue of deforestation by implementing measures to reduce the commercial cutting down of trees for purposes such as timber production and wood extraction. This is performed to conserve forests, protect biodiversity, prevent deforestation, mitigate climate change, and promote sustainable forest management. It involves implementing legal and regulatory measures, establishing protected areas or conservation zones, promoting alternative livelihoods for communities that depend on logging, and encouraging responsible forestry practices that prioritize ecological sustainability [88].

On the other hand, national urban greening policies regulate urban greening rates in the form of laws. It involves the creation and maintenance of parks, gardens, street trees, green roofs, vertical gardens, and planting trees around residential areas [89]. Urban greening aims to enhance the quality of urban environments by providing numerous

benefits. These include reducing air and noise pollution, mitigating the urban heat island effect, improving mental and physical well-being, promoting biodiversity, and creating more pleasant and livable urban spaces. These policies have been effectively implemented in the region, resulting in minimal negative impact of destructive human activities on vegetation greenness.

### 4.3. Sources of Uncertainty

The RESTREND method used in our study is a widely accepted approach for analyzing the driving forces of NDVI dynamics. It effectively distinguishes between the impacts of climatic factors and human activity on ecological restoration. However, it is important to note that the RESTREND method assumes an instantaneous relationship between climate and vegetation, neglecting time-lag effects. The presence of severe human interference or localized redistribution of water and heat can challenge this assumption, leading to unrealistic internal assumptions [74]. Furthermore, the observation of time-lag effects between the NDVI and climate also challenges the internal assumptions and may potentially affect the results of our study [90]. Incorporating time-lag effects into the regression models could be a valuable improvement for the RESTREND method in future research.

In our study, we observed that 4.9% of the grids in the regression models did not reach a significant regression level. These grids were primarily located in urban and agricultural areas, which contributed to the increased uncertainty in our study. One potential solution to address this issue is to use a global regression model (GRM) instead of the local regression model (LRM) that we utilized in this study. The GRM assumes that all grids in a dataset follow the same mathematical relationship and aims to establish a single equation to describe the relationships between variables for the entire dataset. This approach may be more suitable for datasets with overall trends, thus avoiding the issue of nonsignificance encountered in local regression models. However, GRMs have been criticized for neglecting local habitat heterogeneity [91]. On the other hand, LRMs offer more flexibility as they create regression models for each grid in the region. This makes them better suited for situations where the data exhibit local differences, heterogeneity, or nonlinear relationships. The choice between the GRM and the LRM depends on the research question and data characteristics. Comparing and selecting between these two optimization methods can be a potential avenue for future research.

In the exploration of human activity dimensions, we discovered moderately correlated factors, but did not find strongly or very strongly correlated factors. The following question arises: is it meaningful to discuss moderate correlation? Moderate correlation can indeed have significance in certain contexts. While it may not be as strong or direct as strong correlation, moderate correlation can still provide valuable information and clues in research and analysis. Firstly, moderate correlations can offer preliminary clues or directions, helping us understand the problem and lay the foundation for further investigation. Secondly, in complex systems where multiple factors influence phenomena, moderate correlation can provide insights into the system's internal workings. Even if the correlation between a variable and the target variable is weak, it can still play a role in the overall operation of the system. Finally, in multivariate analysis and predictive model optimization, moderate correlation can be important for the overall model, as variables may interact or have common influences. Moderately correlated variables might exhibit stronger predictive ability when combined with other variables. Therefore, moderate correlation can be meaningful in exploratory research, multivariate analysis, predictive model optimization, and understanding complex systems. Further research can focus on exploring the comprehensive impact of moderately correlated variables.

### 4.4. Implications and Future Directions

The findings of our study have the potential to contribute to our understanding of the drivers of ecological restoration in land degradation regions and aid in identifying areas undergoing human-induced changes. Our research focuses on county-level units

rather than natural geographical units because county-level units are the fundamental units of social and economic activities. This study provides insights into the complex interactions between climate, human activities, and vegetation dynamics in semi-humid forest environments. These insights can inform policy and practice in the restoration and sustainable management of these ecosystems not only in the Loess Plateau but also in other regions. Based on our research, we suggest focusing on four aspects of content in the future:

(1) Further investigation into the comprehensive role of socioeconomic factors is crucial. Previous studies have provided limited consideration of these factors and their complex interactions. Future research should aim to deepen our understanding of how human activities, including land-use changes, population density, and economic development, influence vegetation greening at the county scale;

(2) Conducting long-term monitoring and analysis at the county scale would offer a comprehensive understanding of vegetation dynamics and the factors driving them. This approach would enable us to identify trends in vegetation greening and detect any temporal changes or variations in the driving forces;

(3) To obtain a more comprehensive understanding of the factors driving vegetation greening, future research could conduct comparative studies across counties on the Loess Plateau. This approach would allow for the analysis of ecological initiatives, land management practices, and their respective impacts on vegetation greening;

(4) Expanding the research scope to include a broader range of regions outside the Loess Plateau would enhance the generalizability of the findings and provide a more comprehensive understanding of the driving forces behind vegetation greening at the county scale.

## 5. Conclusions

This study effectively disentangled the effects of climate and human activity on vegetation greening using the RESTREND method. Our study shows that the temperature–NDVI relationship is more suitable for establishing regression equations than the precipitation–NDVI relationship in our forest region. Agricultural practices and afforestation significantly promoted vegetation recovery, whereas deforestation and urbanization exhibited negligible impacts on vegetation dynamics. The contribution of human activity outweighed the impact of climate change on vegetation greening. This means that significant progress toward ecological restoration has been achieved with the assistance of human effort in the Loess Plateau. However, future challenges lie in the need to strengthen natural conservation efforts and gradually transition from human-driven restoration processes to those driven by natural forces. This transition will facilitate the more effective utilization of climatic factors as the dominant drivers of ecological restoration, leading to a more sustainable ecological balance (climax theory). Only in this manner can they align with nature-based solutions. Further investigation of the role of socioeconomic factors, long-term monitoring, comparative studies across counties and regions, and exploring more driving models for ecological restoration are important for advancing our understanding of ecological restoration. These efforts will help identify the best restoration paradigm and contribute to more effective and sustainable restoration practices.

**Author Contributions:** Conceptualization, C.K., J.H., S.D. and G.L.; methodology, C.K.; software, C.K. and G.L.; validation, C.K. and G.L.; data curation, C.K. and G.L.; writing—original draft preparation, C.K. and G.L.; writing—review and editing, C.K., J.H., S.D. and G.L.; visualization, C.K. and G.L.; supervision, J.H. and G.L.; funding acquisition, G.L., J.H. and S.D. All authors have read and agreed to the published version of the manuscript.

**Funding:** This research was funded by the joint Funds of the National Natural Science Foundation of China [U2243225], National Natural Science Foundation of China grant number [31971488] and the National Key Research and Development Program of China grant number [2017YFC0504601].

**Data Availability Statement:** Data are contained within the article.

**Acknowledgments:** We are grateful to the three reviewers for their invaluable suggestions on our manuscript, which have significantly enhanced the quality of our work. We appreciate the language editor's modifications to the language and format of the manuscript. We appreciate the assistance of Yueni Zhang, Guan Liu, Ying Liu, and Dongyang Xiong in data collection, organization, and methodology.

**Conflicts of Interest:** The authors declare no conflicts of interest. The founding sponsors had no role in the design of this study; in the collection, analyses, or interpretation of data; in the writing of the manuscript; or in the decision to publish the results.

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
