# Peer review of "Exploring the Driving Forces of Vegetation Greening on the Loess Plateau at the County Scale"

_forests, doi:10.3390/f15030486_

Round 1

Reviewer 1 Report (Previous Reviewer 4)

Comments and Suggestions for Authors

Dear Authors,

I have had the opportunity to review the revised version of your manuscript titled "Exploring the Driving Forces of Vegetation Greening on the Loess Plateau at the County Scale." I am pleased to note the significant improvements made in this version, addressing many of the earlier concerns. The manuscript is now much more robust and clearer in its presentation. However, I have a few additional comments that could further enhance the paper.

  1. Remote Sensing Section:

    • Suggestion: At the beginning of the Remote Sensing section, consider first mentioning the use of SPOT satellite imagery. Following this, you can then explain the various indices used in your study. This sequence would provide a logical flow, helping readers to understand the context before delving into the specifics of the indices.
  2. Figure 4 – RESTREND Analysis:

    • Observation: The RESTREND analysis presented in Figure 4 is well described. However, there seems to be a gap in the explanation regarding how the driving forces for human activities are measured and incorporated into the analysis.
    • Recommendation: Please include a detailed explanation of the methodology used to quantify and integrate the impact of human activities. This would provide a complete picture of how both natural and anthropogenic factors are being assessed in your study.

These comments are not critical but are intended to refine the manuscript further. The additions suggested would contribute to the overall clarity and comprehensiveness of your research.

Thank you for considering my feedback. I look forward to the possibility of seeing this enhanced version of your manuscript.

Author Response

Dear Editors and Reviewers:

Thank you for your letter and for the reviewers’ comments concerning our manuscript entitled “Exploring the Driving Forces of Vegetation Greening on the Loess Plateau at the County Scale” (ID: forests- 2851949). Those comments are all valuable and very helpful for revising and improving our paper, as well as the important guiding significance to our researches. We have studied comments carefully and have made correction which we hope meet with approval. Revised portion are marked in red in the text. Besides, we have polished the English language to meet the requirements for publication. The main corrections in the paper and the responds to your comments are as flowing:

  1. Remote Sensing Section:
    • Suggestion: At the beginning of the Remote Sensing section, consider first mentioning the use of SPOT satellite imagery. Following this, you can then explain the various indices used in your study. This sequence would provide a logical flow, helping readers to understand the context before delving into the specifics of the indices.

Response: At the beginning of the section 2.2.1. Remote sensing data, we have provided additional details about the use of SPOT satellite imagery: “The SPOT satellite (Satellite Pour I' Observation de la Terre), developed by the Centre National D'Etudes Spatiales (CNES), is an earth observation satellite used for remote sensing. Since 1986, a series of SPOT satellites (SPOT 1-7) have been launched, providing continuous remote sensing data for various application fields. The SPOT satellite can capture images in the visible light band as well as capturing multi-spectral information, including infrared bands. This makes the SPOT satellite particularly valuable for analyzing vegetation, land cover, water bodies, soil, and other related studies. In vegetation research, SPOT satellite images can be used to obtain vegetation coverage information, monitor changes in vegetation, assess vegetation health, and more. Due to its high spatial resolution and multispectral capabilities, the SPOT satellite is considered an important data source in remote sensing research.”

  1. Figure 4 – RESTREND Analysis:
    • Observation: The RESTREND analysis presented in Figure 4 is well described. However, there seems to be a gap in the explanation regarding how the driving forces for human activities are measured and incorporated into the analysis.
    • Recommendation: Please include a detailed explanation of the methodology used to quantify and integrate the impact of human activities. This would provide a complete picture of how both natural and anthropogenic factors are being assessed in your study.

Response: We have detailed the methods used to quantify and integrate the impacts of human activities in Section 2.3.2. The workflow for statistical analysis: “Finally, we investigated the impact of human activities on regression residuals. Here, we calculated the correlation between NDVI and each socioeconomic variable using the Pearson correlation coefficient. The formula used for these calculations is as follows:

Ri =cor(Trend, social[[i]], method = "pearson")                         (6)

where, social[[i]] is the ith human disturbance factor (afforestation intensity, deforestation intensity, agricultural intensity, village intensity, and road intensity), and Ri is the correlation coefficient between Trend and the social[[i]] . The significance level was set at p < 0.05.” We also modified Figure 4.

We gratefully acknowledge your permission to submit a revised version of the manuscript. Thank you again for your tremendous patience and hard work on our manuscript. We look forward to your feedback on the revised manuscript.

Sincerely,

Guoqing Li

Institution and address: Xinong Rd. 26, Institute of Soil and Water Conservation, Yangling, Shaanxi 712100, China.

Telephone: +86 15091543105

E-mail: liguoqing@nwsuaf.edu.cn

Reviewer 2 Report (New Reviewer)

Comments and Suggestions for Authors

Reviewer response to the authors.

This paper presented a study of evaluating and analysing ecological restoration and driving factors from 1999 to 2019. The results showed the impacts of vegetation greenness and recovery. This study may be helpful for government policy and sustainable land management. However, there are a few shortcomings that need attention:

1.     In the abstract, the authors should provide more information on the method part.

2.     In remote sensing data, SPOT vegetation's spatial resolution is 1x1 km2. Is it suitable for your study? Did you conduct resampling?

3.     Did the authors use Max. Min. Mean or Median climate data for each of 21 years?

4.     In Figure 3, you showed mean NDVI. Which sensor do you use, and which temporal data did you prepare?  

5.     Socioeconomic factors are still not clear to me. How did you use the data from 1999 to 2019 due to land change over time?

6.     It seems like you used the residual trend analysis as one of your objectives. As a result, in section 3.1, I was unclear about how you produced the map using Max_NDVI from 1999 to 2019. Also, how do you implement the data over 21, as you always mentioned the trends?

7.     Where is the result of the trend of changes? The trend change should be a line graph or not? You need to provide more information. 

8.     In the discussion, I would success you to improve it, particularly in line 389.

9.     In the conclusions and outlook, what are the key points and suggestions for future study?   

Comments on the Quality of English Language

Minor editing of the English language required

Author Response

Dear Editors and Reviewers:

Thank you for your letter and for the reviewers’ comments concerning our manuscript entitled “Exploring the Driving Forces of Vegetation Greening on the Loess Plateau at the County Scale” (ID: forests- 2851949). Those comments are all valuable and very helpful for revising and improving our paper, as well as the important guiding significance to our researches. We have studied comments carefully and have made correction which we hope meet with approval. Revised portion are marked in red in the text. Besides, we have polished the English language to meet the requirements for publication. The main corrections in the paper and the responds to your comments are as flowing:

  1. In the abstract, the authors should provide more information on the method part.

Response: In the abstract, we have provided information on the method part: “Five of socioeconomic variables were considered, including afforestation intensity, deforestation intensity, agricultural intensity, village intensity, and road intensity layers, to characterize the impact of afforestation, agriculture, and urbanization policies. The RESTREND (residual trends) method was employed to assess the relative importance of climate and human activities on vegetation dynamics.”

  1. In remote sensing data, SPOT vegetation's spatial resolution is 1x1 km2. Is it suitable for your study? Did you conduct resampling?

Response: We did not resample the SPOT remote sensing data. “This dataset is widely used for monitoring vegetation changes at various scales and is particularly valuable for studying vegetation change trends in larger areas. While the 1km resolution may not capture microscopic changes in vegetation, it is sufficient for identifying and analyzing the main patterns of vegetation change, especially at the county scale.”

  1. Did the authors use Max. Min. Mean or Median climate data for each of 21 years?

Response: We have made the following additions to the use of climate data: (1) in section 2.2.2. Climate data: “Furthermore, the meteorological data obtained for this study are based on monthly observations.” (2) In section 2.3.2. The workflow for statistical analysis, the 21-year growing season total precipitation and mean temperature data from the meteorological stations were used.

  1. In Figure 3, you showed mean NDVI. Which sensor do you use, and which temporal data did you prepare?  

Response: We shown the average NDVI in Figure 3, mainly to show the overall status of vegetation in Huanglong County from 1999 to 2019. The data used is still the NDVI data set mentioned in Section 2.2.1. The current title of Figure 3 is “Figure 3. Map of annually average of maximum NDVI (1999-2019) and five socioeconomic factors representing human activities.”

  1. Socioeconomic factors are still not clear to me. How did you use the data from 1999 to 2019 due to land change over time?

Response: Regarding the generation of socioeconomic data layers, we have made the following additions in section 2.2.3. Socioeconomic factors: “In the generation of afforestation intensity and deforestation intensity data, the comparison of land use conditions in 1999 and 2019 was conducted to capture the changing trends in the region over the 21-year period. While this approach does not provide detailed annual change information, it offers insights into the long-term trends and overall directions of change in the area. Regarding the data on agricultural intensity, village intensity, and road intensity, the selection of data from 2019 allows for the assessment of the current impact of agricultural activities and urbanization levels on vegetation restoration in the region.”

  1. It seems like you used the residual trend analysis as one of your objectives. As a result, in section 3.1, I was unclear about how you produced the map using Max_NDVI from 1999 to 2019. Also, how do you implement the data over 21, as you always mentioned the trends?

Response: We provide information in section 3.1. Stepwise regression and residual trend analysis: “We extracted grid values and climate data from the 21-year raster layer at corresponding locations. Each NDVI grid value was considered as the dependent variable, while the climate data was treated as the independent variable in establishing a regression relationship. This process was repeated for 2811 grids. Table 1 provides a summary of the regression results, indicating the established correlation between NDVI and climate factors in our study. The table shows that 95% of the grids exhibited good predictive performance.”

  1. Where is the result of the trend of changes? The trend change should be a line graph or not? You need to provide more information. 

Response: We have supplemented the information in section 2.3.2. The workflow for statistical analysis: "Thirdly, a linear regression equation was employed to calculate the 21-year regression residuals for each grid and corresponding year. The formula used for this calculation is as follows:

Tredi = coef(lm(Resdi~year))                                (3)

where, Tredi represents the residual trend (regression slope) for grid i, and  marks the duration from 1999 to 2019. The coef represents a function for extracting regression slopes. When Tredi > 0 indicated that human activities had a positive effect on ecological restoration, while  Tredi < 0 indicated a negative impact.".

Secondly, we also provide information in section 3.1. Stepwise regression and residual trend analysis: “We visualized the spatial distribution of the residual trend (regression slope) for 2811 grids using a raster layer, as shown in Figure 5E. In the figure, a positive value represents a positive trend. This two-dimensional representation allows readers to perceive the changes in trends more effectively compared to a line chart with a single value. Figure 5F provides a statistical indicator that helps us understand the frequency of the residual trend in the regression equation.”

  1. In the discussion, I would success you to improve it, particularly in line 389.

Response: We have improved the discussion section and made the following changes to the original content of line 389: “However, the presence of time-lags may introduce uncertainty, as mentioned in section 4.3 of the paper. This uncertainty has implications for the analysis and interpretation of the data.”

  1. In the conclusions and outlook, what are the key points and suggestions for future study? 

Response: We added section 4.4 Implications and future directions and made the following recommendations for future research: “(1) Further investigation of the role of socioeconomic factors: Given the limited consideration of socioeconomic factors in previous studies, future research could delve deeper into understanding the influence of human activities, such as land use changes, population density, and economic development, on vegetation greening at the county scale. (2)       Long-term monitoring and analysis: Conducting long-term monitoring and analysis of vegetation dynamics and driving factors at the county scale would provide a more comprehensive understanding of vegetation greening trends and help identify any temporal changes or variations in driving forces. (3) Comparative studies across counties: To gain a more comprehensive understanding of the driving forces behind vegetation greening, future research could compare and analyze multiple counties on the Loess Plateau. This comparative approach would help identify variations in ecological initiatives, land management practices, and their impacts on vegetation greening. (4) Inclusion of more regions: Expanding the research scope by including a broader range of regions outside the Loess Plateau would enhance the generalizability of the findings and provide a more holistic understanding of the driving forces of vegetation greening at the county scale.”

  1. Minor editing of the English language required

Response: We have made an overall improvement to the language expression of the manuscript, hoping to make it more standardized and easier for readers to read.

We gratefully acknowledge your permission to submit a revised version of the manuscript. Thank you again for your tremendous patience and hard work on our manuscript. We look forward to your feedback on the revised manuscript.

Sincerely,

Guoqing Li

Institution and address: Xinong Rd. 26, Institute of Soil and Water Conservation, Yangling, Shaanxi 712100, China.

Telephone: +86 15091543105

E-mail: liguoqing@nwsuaf.edu.cn

Reviewer 3 Report (New Reviewer)

Comments and Suggestions for Authors

The manuscript by Kong et al. considers the influence of different factors (including human activity, precipitation, temperature) on vegetation during twenty years. The theme is very important for development of environmental management. However, I have some questions and comments.

(1)    Were the parametrization of equation (1) and calculation of residual NDVI (equation 2) based on same dataset of NDVI? Why this residual NDVI can be used for calculation of the rate of climatic contribution to the NDVI (equation 4) and the rate of human activity contribution to the NDVI (equation 5)? The method which is described in section 2.3 is not clear.

(2)    The correlations in Table 2 are weak and cannot be used for estimation of influence of factors on vegetation dynamic.

(3)    Please add dynamics of NDVI, Resd.slope, Afforestation, Agriculture, Population, Deforestation, Road for 1999-2019 years.

(4)    Please, add in maps spatial resolution bars.

Author Response

Dear Editors and Reviewers:

Thank you for your letter and for the reviewers’ comments concerning our manuscript entitled “Exploring the Driving Forces of Vegetation Greening on the Loess Plateau at the County Scale” (ID: forests- 2851949). Those comments are all valuable and very helpful for revising and improving our paper, as well as the important guiding significance to our researches. We have studied comments carefully and have made correction which we hope meet with approval. Revised portion are marked in red in the text. Besides, we have polished the English language to meet the requirements for publication. The main corrections in the paper and the responds to your comments are as flowing:

  1. Were the parametrization of equation (1) and calculation of residual NDVI (equation 2) based on same dataset of NDVI? Why this residual NDVI can be used for calculation of the rate of climatic contribution to the NDVI (equation 4) and the rate of human activity contribution to the NDVI (equation 5)? The method which is described in section 2.3 is not clear.

Response: Firstly, the parameterization of equation (1) and the calculation of residual NDVI (equation 2) are based on the same NDVI data set, and we supplement the explanation at equation 2: “where,  represents the 21-year sequence of NDVI data for grid i".

Secondly, we have made the following additions to the method part: "The method described in this study is based on two assumptions. Firstly, it establishes a relationship between climate factors and NDVI to obtain predicted values, which represent the variability directly attributed to climate factors. Secondly, the residual is calculated as the difference between actual observed values and predicted values, capturing the variation that cannot be explained by climate alone. Typically, this unexplained variation is attributed to human activities. It's important to note that the essence of RESTREND thought is binary division and does not account for the influence of random processes. By utilizing predicted values and residuals, it becomes possible to effectively distinguish the impact of climate change and human activities on NDVI changes.".

  1. The correlations in Table 2 are weak and cannot be used for estimation of influence of factors on vegetation dynamic.

Response: In this study, we utilized residual trend analysis to generate a raster layer of residual trends and performed linear regression by integrating five socioeconomic data layers. However, the overall correlation coefficient is relatively low, suggesting that estimating the impact of human activities on vegetation dynamics may lack precision. This limitation primarily arises from the constrained quantifiability of socioeconomic factors, thus restricting our analysis to these five variables. Such uncertainty underscores the complexity of factors influencing our findings, some of which may not be fully captured.

We also make recommendations in Section 4.4. Implications and future directions: “Based on our research, we suggest focusing on four aspects of content in the future: (1) Further investigation of the role of socioeconomic factors: Given the limited consideration of socioeconomic factors in previous studies, future research could delve deeper into understanding the influence of human activities, such as land use changes, population density, and economic development, on vegetation greening at the county scale.”

  1. Please add dynamics of NDVI, Resd.slope, Afforestation, Agriculture, Population, Deforestation, Road for 1999-2019 years.

Response: The dynamics of NDVI can be seen from Figure 5E, which indicates a trend change in NDVI caused by human interference after removing climate effects. However, Afforestation, Agriculture, Population, Deforestation, and Road are not dynamically changing, but rather a sustained force.

In Figure 3, we have expressed the NDVI, afforestation, agriculture, population, deforestation, and road layers. Regarding the generation of socioeconomic data layers, we have made the following additions in section 2.2.3. Socioeconomic factors: “In the generation of afforestation intensity and deforestation intensity data, the comparison of land use conditions in 1999 and 2019 was conducted to capture the changing trends in the region over the 21-year period. While this approach does not provide detailed annual change information, it offers insights into the long-term trends and overall directions of change in the area. Regarding the data on agricultural intensity, village intensity, and road intensity, the selection of data from 2019 allows for the assessment of the current impact of agricultural activities and urbanization levels on vegetation restoration in the region.”

  1. Please, add in maps spatial resolution bars.

Response: Thanks for your advice. However, we believe that increasing the scale is unnecessary because the latitude and longitude are already included in the correlation figures. With this information, readers can determine the specific location and size of the study area, thereby reflecting spatial resolution. Therefore, adding additional spatial resolution bars may appear redundant and may visually clutter the figures.

We gratefully acknowledge your permission to submit a revised version of the manuscript. Thank you again for your tremendous patience and hard work on our manuscript. We look forward to your feedback on the revised manuscript.

Sincerely,

Guoqing Li

Institution and address: Xinong Rd. 26, Institute of Soil and Water Conservation, Yangling, Shaanxi 712100, China.

Telephone: +86 15091543105

E-mail: liguoqing@nwsuaf.edu.cn

Round 2

Reviewer 2 Report (New Reviewer)

Comments and Suggestions for Authors

All comments have been addressed by the authors.  

Author Response

Thank you again for your hard work and valuable suggestions on our manuscript, which has greatly improved the quality of our work.

Reviewer 3 Report (New Reviewer)

Comments and Suggestions for Authors

Authors minimally considered my comments. However, I suppose that the investigation has two important problems. Firstly, the difference between the approximation model based on experimental data, and the same experimental data were used to indicate the influence of a climatic factor. This difference indicates an error of the model. I am not sure that this error can be correct indicator of climatic factor. Secondly, low correlation coefficients between indicators in Table 2 do not seem to be very informative and cannot be basis of valid conclusions.

Author Response

Authors minimally considered my comments. However, I suppose that the investigation has two important problems. Firstly, the difference between the approximation model based on experimental data, and the same experimental data were used to indicate the influence of a climatic factor. This difference indicates an error of the model. I am not sure that this error can be correct indicator of climatic factor . 
Response: Thank you for your comment. It seems that there may be a bias in your understanding of the RESDTRED method. The regression error (Tresd in Eq.2) should be considered as an indicator of human factors rather than climatic factors.
We have provided a detailed explanation of the underlying philosophical concepts of the RESDTRED method in section 2.3.1 of the methodology. “The method described in this study is based on two assumptions. Firstly, it establishes a relationship between climate factors and NDVI to obtain predicted values, which represent the variability directly attributed to climate factors. Secondly, the residual is calculated as the difference between actual observed values and predicted values, capturing the variation that cannot be explained by climate alone. Typically, this unexplained variation is attributed to human activities. It's important to note that the essence of RESTREND thought is binary division and does not account for the influence of random processes. By utilizing predicted values and residuals, it becomes possible to effectively distinguish the impact of climate change and human activities on NDVI changes.”

Secondly, low correlation coefficients between indicators in Table 2 do not seem to be very informative and cannot be basis of valid conclusions.
Response: That’s a good question. Regarding the testing of the 5 hypotheses for the 5 social factors using correlation coefficients, we have incorporated the following criteria in the method section: “The value range of R, the correlation coefficient, is between -1 and 1. Positive values indicate a positive correlation, while negative values indicate a negative correlation. The absolute value of the correlation coefficient is used to determine the strength of the correlation. In general, a range of 0-0.1 indicates a negligible correlation, a range of 0.1-0.39 indicates a weak correlation, a range of 0.4-0.69 indicates a moderate correlation, a range of 0.7-0.89 indicates a strong correlation, and a range of 0.9-1 indicates a very strong correlation [78]. Here, the threshold of modal correlation was used to detect the important social factor that affects residual trends. Variables with R greater than this threshold will be considered the main factor affecting residual trends.”

In the uncertainty section, we have included a discussion on the ecological significance of moderate correlations: "In the exploration of human activity dimensions, we discovered moderately strong correlated factors, but did not find strongly or very strongly correlated factors. The question arises: is it meaningful to discuss moderate correlation? Moderate correlation can indeed have significance in certain contexts. While it may not be as strong or direct as strong correlation, moderate correlation can still provide valuable information and clues in research and analysis. Firstly, moderate correlations can offer preliminary clues or directions, helping us understand the problem and lay the foundation for further investigation. Secondly, in complex systems where multiple factors influence phenomena, moderate correlation can provide insights into the system's internal workings. Even if the correlation between a variable and the target variable is weak, it can still play a role in the overall operation of the system. Finally, in multivariate analysis and predictive model optimization, moderate correlation can be important for the overall model, as variables may interact or have common influences. Moderately correlated variables might exhibit stronger predictive ability when combined with other variables. Therefore, moderate correlation can be meaningful in exploratory research, multivariate analysis, predictive model optimization, and understanding complex systems. Further research can focus on exploring the comprehensive impact of weakly correlated variables.”

Thank you again for your hard work and valuable suggestions on our manuscript, which has greatly improved the quality of our work.

This manuscript is a resubmission of an earlier submission. The following is a list of the peer review reports and author responses from that submission.

Round 1

Reviewer 1 Report

Comments and Suggestions for Authors

The paper titled "Exploring the Driving Forces of Vegetation Greening on the Loess Plateau at the County Scale " examines the greening of vegetation on the Loess Plateau, focusing especially on the Normalized Difference Vegetation Index (NDVI) as a crucial metric. While the authors have gathered and examined substantial data, the paper's innovative methods and its unique contributions remain unclear. Additionally, the paper would benefit from a more detailed discussion of the results and their implications, improved integration of global literature (as it overly concentrates on studies conducted in Asia, but this is a journal with a global scope), and enhanced data presentation. There's also a notable gap in the depth of discussion regarding the connection between NDVI, satellite imagery, and vegetation, which the authors claim is central to the study. For instance, the link between agriculture and NDVI is mentioned multiple times, but its nature remains ambiguous. I suggest adding some background with information about why NDVI is appropriate to monitor vegetation and agriculture:

- “High-Resolution NDVI from Planet’s Constellation of Earth Observing Nano-Satellites: A New Data Source for Precision Agriculture”. https://doi.org/10.3390/rs8090768

- “Effect of Missing Vines on Total Leaf Area Determined by NDVI Calculated from Sentinel Satellite Data: Progressive Vine Removal Experiments”. https://doi.org/10.3390/app10103612.

- “Agriculture Phenology Monitoring Using NDVI Time Series Based on Remote Sensing Satellites: A Case Study of Guangdong, China” https://doi.org/10.3103/S1060992X19030093.

- “The influence of land cover-related changes on the NDVI-based satellite agricultural drought indices” https://doi.org/10.1109/IGARSS.2014.6946868

In the materials and methods section, the reason behind selecting the NDVI dataset is not sufficiently explained, and more information about the sensors used for data collection is necessary. The explanation of the RESTREND methodology and the application of NDVI data is extensive, yet more clarity on why this method was chosen over others, and its advantages, would clarify the paper. Occasionally, the connection between the method and the data is not explicit, as the authors fail to adequately explain the implications of using NDVI over RGB images or other vegetation indices.

The results section is organized and descriptive, and the figures are clear.

The discussion section is informative but lacks in-depth engagement with existing research. Adding a comparative analysis with similar studies would enhance its depth. The study's findings' implications for policy and practice are only briefly touched upon. A more comprehensive examination of how these results might guide ecological restoration efforts on the Loess Plateau is necessary.

The conclusion concisely presents the main findings but could emphasize the study's novel contributions more effectively, as these are currently not clear.

Comments on the Quality of English Language

The manuscript's English language usage requires a review, as evidenced by various instances of mistakes and awkward phrasing. For example, on line 33, the term "domain tools" is used, which might be more appropriately expressed as "main tools." In line 56, there seems to be a missing "is" in the phrase “changes in vegetation greenness predominantly.” Similarly, line 59 “counties have completed social and economic” appears to incorrectly use "completed" instead of the more fitting “complete.” Additionally, line 90 “trend of vegetation change trends” is redundant. These are just a few instances highlighting the need for a language revision in the manuscript.

Reviewer 2 Report

Comments and Suggestions for Authors

The paper explores NDVI dataset and employs the RESTREND method to investigate ecological restoration efforts in the Loess Plateau region from 1999 to 2019. This paper is logically organized. I believe this paper could be published after the following concerns have been addressed:

·      Define the acronym when first used

·    Lines 38-41: Long sentence needs to be rewritten.

·   The RESTREND method is not mentioned in the introduction section. I suggest its inclusion for enhanced clarity and context. Additionally, it is recommended to highlight other prevalent methods used in assessing and analyzing ecological restoration.

· Overall, the introduction is still insufficient; it does not provide foundations to convince the reader that the research is relevant. I suggest emphasizing the paper's originality.

·   In the study area subsection, kindly include specific details about the types of ecosystems present, biodiversity, and any unique ecological features.

·Figure 1 requires improvement. It should provide a clearer representation of the location in relation to nearby countries and add the study area's borders.

·    Line 97: What is the NDVI's spatial resolution used in this research? and from which sensor?

·    Line 98: Justify the choice of the maximum synthesis method

·    Line 100: At which time scale? Daily, monthly, or yearly?

·   Line 133: For NDVI you acquired 21 rasters representing values from 1999 to 2021. However, there were only 19 employed. Can you please explain this discrepancy?

·   I suggest inserting a methodology flowchart of data and analysis used in this research.

·    Which coordinate projection system did you use for all spatial maps?

·  Overall, the approach looks like describes the results obtained. Thus, critical analysis is required.

·     The discussion and conclusion sections need to be more compelling.

Reviewer 3 Report

Comments and Suggestions for Authors

The manuscript entitled “Exploring the Driving Forces of Vegetation Greening on the Loess Plateau at the County Scale” investigates the trends of vegetation densening and recovery over a county in the region of Loess Plateau and for a period of 21 years. Furthermore, it attempts to interpret the observed trends using climatic and anthropogenic variables. The main findings of the study is that vegetation is indeed increasing in the study period and to a great extend this is the result of anthropogenic actions rather than the result of climatic trends, which of course have its own impact. Furthermore, according to the results presented temperature appears to be more effective than precipitation in promoting vegetation increase. The subject of the study is interesting but I believe the manuscript needs to be improved before it is ready for publication.

The introduction fails to put the study into a general and global context of land use and vegetation recovery trends. It wasn’t easy to see the purpose of the study and what new will it convey. Furthermore, I would prefer to see a clear aim and a set of objectives as the last paragraph in the introduction.  

 My main concern about this manuscript is in the methods section. While I do not question the scientific soundness of the methods employed, I believe they are poorly described. For instance, the authors use NDVI as the proxy of vegetation recovery, but it is not clear from which sensor this NDVI was derived. Although the source is referenced the authors should provide more information on the main dataset of their study. What this NDVI refers to? Is it a mean annual value or monthly values or daily values?  What is the spatial resolution of the NDVI and the subsequent spatial data?   

The authors used five socioeconomic layers which were generated using a Kernel Density Estimation Function but what was the vector data used to generate those layers. Roads and Villages are clear but how Afforestation and Deforestation was calculated? The authors should provide more information on how their data were generated so one can repeat the process in another area.

Which method of spatial interpolation did the authors employ to convert the climatic data into a raster layer? It is not clear in the text

In lines 125-126 the authors state that raster NDVI data were converted to vector data but based on the description it seems that the raster data were simply converted into a two dimensions matrix were each row corresponds to a cell in the original raster and each column is the NDVI value of a particular year.  Is this correct or the authors have created a vector data set (Polygons or points) where each polygon or point corresponds to a cell and it is associated by an attributed table with 19 attributes? And why the authors use 19 NDVI values and not 21 as stated in line 97?

The results are also poorly presented. It is not clear which parameters have a significant impact on vegetation trends. No measurement of significance is presented apart from the distribution of R2  which is not informative. For instance in lines 176-177 the authors state that the positive residuals indicate a positive human impact. How this conclusion is drawn? It needs to be explained.  The authors should bear in mind that not all readers are familiar with the employed methods so when the results are presented they have to be clear and convey a clear message.

In the first paragraph of the discussion the authors claim that the more significant impact of temperature to vegetation recovery is probably due to the geographical region that the study took place. Personally I believe that it is due to the limitation of the study which does not consider time-lags. Although the authors acknowledge this limitation  I believe it would be useful to mention it here as well. The paragraph in lines 224-240 is in my view very dubious. In this paragraph the authors claim that global warming may benefit vegetation recovery by increasing the growing season and making plants to be benefited by a more efficient water use. The study was conducted in an area of a relatively low precipitation which is concentrated within a period of 3 months. Although this period corresponds to the growing season a claim that increasing temperature is benefitting vegetation needs to be justified better.  With the exception of tropical regions vegetation is recovering almost everywhere in the world due primarily to changes in socioeconomic factors and a reduction of the human presence in natural areas. This allows the vegetation to recover and in my view temperature increase and vegetation recovery is simply two processes that co-exist without immediate relationship between them. It is very risky to claim that global warming is benefitting ecological restoration and if the authors insist on that they have to justify it better.

In lines  256-280 the authors discuss the observed positive effect of agricultural activity on vegetation greening and they suggest that the increase in use of pesticides and fertilisers is rather positive. Although the authors in lines 272-280 express their concerns on this phenomenon I believe this paragraph has to be rewritten and personally if such pattern does really exist the interpretation probably lies in some land abandonment or set aside or changes in cultivation patterns.  

I believe the manuscript is interesting and it has potential but in my view, it has to be revised considerably. All my comments aim only in the improvement of the manuscript, its more comprehensive presentation and the improvement of its readability. The authors are of course free to accept or decline any of my comments.

Comments on the Quality of English Language

The manuscript reads relatively well but I believe it will be benefitted if it is edited by a native English speaker

Reviewer 4 Report

Comments and Suggestions for Authors

Dear Authors,

I have had the opportunity to thoroughly read and analyze your manuscript titled "Exploring the Driving Forces of Vegetation Greening on the Loess Plateau at the County Scale." I appreciate the effort and research you have put into this significant topic. However, I have a few suggestions and comments that I believe could enhance the quality and impact of your paper.

  1. Introduction and Literature Review:

    • Currently, your introduction and literature review are mainly focused on studies related to the Loess Plateau. I recommend broadening this section to include a more diverse range of studies on ecological restoration. This expansion will provide a comprehensive background and situate your study within a global context.
  2. Role of Temperature and Climatic Changes:

    • I observed that the paper includes graphs related to temperature and climatic changes. However, these graphs lack clear and detailed explanations. It would be beneficial to elaborate on how temperature and other climatic factors are influencing vegetation greening, making the connection between climate change and ecological restoration more explicit.
  3. Methodology:

    • The methodology section of the paper is described quite simplistically and lacks clarity. I advise expanding this section to detail your methodology, explaining the rationale behind your choice of methods and how they contribute to the reliability of your findings.
  4. Contribution to Ecological Restoration:

    • From the results section, it is not clear how your study contributes to the field of ecological restoration. It would be valuable to highlight the implications of your findings for ecological restoration, discussing how your research adds to existing knowledge and its potential impact on policy and practice.

General Observations:

  • Overall, the paper requires more research and a better description of its components. In its current form, the manuscript may face challenges in being accepted for publication. Enhancing the depth of research and clarity of your paper will significantly improve its potential for publication.

In conclusion, while your study addresses a critical and timely topic, implementing the above suggestions could greatly enhance its contribution to the field of ecological restoration. I look forward to seeing the revised version of your manuscript.
